# Development of a Novel Loop Mediated Isothermal Amplification Assay (LAMP) for the Rapid Detection of Epizootic Haemorrhagic Disease Virus

**DOI:** 10.3390/v13112187

**Published:** 2021-10-29

**Authors:** Paulina Rajko-Nenow, Emma L. A. Howson, Duncan Clark, Natasha Hilton, Aruna Ambagala, Nicholas Svitek, John Flannery, Carrie Batten

**Affiliations:** 1The Pirbright Institute, Ash Road, Pirbright GU24 0NF, UK; emma@optigene.co.uk (E.L.A.H.); john.flannery@ucd.ie (J.F.); 2OptiGene Limited, Blatchford Road, Horsham RH13 5QR, UK; duncan@optigene.co.uk (D.C.); natasha@optigene.co.uk (N.H.); 3National Center for Foreign Animal Disease, Canadian Food Inspection Agency, 1015 Arlington Street, Winnipeg, MB R3G 1Y5, Canada; aruna.ambagala@inspection.gc.ca; 4Internatonal Livestock Research Institute, Nairobi P.O. Box 30709, Kenya; n.svitek@cgiar.org

**Keywords:** RT-LAMP, EHDV, rapid diagnostics, multiplex LAMP, AB 7500 fast instrument

## Abstract

Epizootic haemorragic disease (EHD) is an important disease of white-tailed deer and can cause a bluetongue-like illness in cattle. A definitive diagnosis of EHD relies on molecular assays such as real-time RT-qPCR or conventional PCR. Reverse transcription loop-mediated isothermal amplification (RT-LAMP) is a cost-effective, specific, and sensitive technique that provides an alternative to RT-qPCR. We designed two sets of specific primers targeting segment-9 of the EHD virus genome to enable the detection of western and eastern topotypes, and evaluated their performance in singleplex and multiplex formats using cell culture isolates (n = 43), field specimens (n = 20), and a proficiency panel (n = 10). The limit of detection of the eastern and western RT-LAMP assays was estimated as ~24.36 C_T_ and as ~29.37 C_T_ in relation to real-time RT-qPCR, respectively, indicating a greater sensitivity of the western topotype singleplex RT-LAMP. The sensitivity of the western topotype RT-LAMP assay, relative to the RT-qPCR assay, was 72.2%, indicating that it could be theoretically used to detect viraemic cervines and bovines. For the first time, an RT-LAMP assay was developed for the rapid detection of the EHD virus that could be used as either a field test or high throughput screening tool in established laboratories to control the spread of EHD.

## 1. Introduction

Epizootic haemorrhagic disease (EHD), caused by the EHD virus (EHDV), is an infectious, non-contagious disease that is transmitted by *Culicoides* biting midges. EHD affects wild and domestic ruminants and causes a severe haemorrhagic disease in white-tailed deer with high morbidity and mortality, and is considered one of the most important diseases affecting deer [1,2]. Milder disease and higher survival rates have been reported for other species such as mule deer, black-tailed, deer and pronghorn antelope [3]. In cattle, EHD resembles a bluetongue (BT)-like illness and can cause significant production losses due to decreased milk production, abortions, and malformations. Historically, EHDV-2 (Ibaraki virus) caused large-scale outbreaks of disease in cattle in Japan, in 1959 [4]. More recently, additional serotypes (EHDV-6 and -7), have been reported to cause severe clinical signs of EHD in dairy and beef cattle in countries neighboring the European Union (EU), such as Turkey, Morocco, Tunisia, Algeria, Israel, and Jordan [2]. The European Food Safety Authority Panel on Animal Health and Welfare concluded that the presence of EHD in EU neighboring countries poses a significant risk for the introduction and establishment of this virus in Europe; therefore, EHD was added to the list of notifiable diseases by the World Organization for Animal Health (OIE) [2]. In 2016, EHDV-1 was simultaneously recorded in cattle in Israel [5] and Egypt [6], often co-circulating with other pathogens, including Bluetongue virus (BTV).

EHDV is classified within the genus *Orbivirus* of the family *Reoviridae* and is both genetically and morphologically related to BTV. The EHDV genome comprises ten segments of linear double-stranded (ds) RNA that encode for structural (VP1-VP7) and non-structural proteins (NS1−NS4, and putatively NS5) [7,8,9]. The outer-capsid is made of two structural proteins VP2 and VP5, which are encoded by segment 2 and 6, respectively. The highly variable VP2 protein is the primary determinant of serotype specificity through interactions with neutralizing antibodies; however, VP5 also elicits neutralizing antibodies [10]. Based on a phylogenetic analysis, the EHDV serotypes are classified into four groups (A–D), which have been shown to correspond well with serological properties of the virus, with no cross-neutralization occurring between the groups. However, some confusion regarding the number of serotypes and their nomenclature still exists; historically, eight serotypes were recognized [11], but following the analysis of sequencing data, EHDV-3 was recategorized as EHDV-1 [10]. Most recently novel serotypes such as EHDV-10 in Japan [12] and YNDH/V079/2018 in China [13] has been proposed but not officially recognized. As with BTV, EHDV strains can be categorized based on their geographical origin into “eastern” (i.e., Asia and Australia) and “western” (i.e., Americans, Africa, and the Middle-East) topotypes [8,10] through sequencing of the core protein (VP7) or non-structural genes. However, the eastern topotype strain (TAT2013/02), recently detected in Trinidad and Tobago [14,15], and a novel EHDV isolate (YNDH/V079/2018), detected in China, did not cluster with either the eastern or western topotypes [13].

A definitive diagnosis of EHD relies on specific laboratory-based tests, as EHD clinical signs are indistinguishable from those of BT and they can be similar to clinical signs observed for other diseases such as bovine viral diarrhea, infectious bovine rhinotracheitis, or vesicular stomatitis [2]. In recent years, a number of real-time RT-qPCR assays have been developed to specifically detect EHDV, targeting segment-5 (Seg-5) [16], Seg-9 [17,18,19], or Seg-10 [20] of the viral genome. While RT-qPCR methods are accurate, rapid, and sensitive, they remain largely restricted to laboratory settings. Although RT-qPCR is widely used in developed countries, it is still considered an expensive technique in developing countries where equipment, reagents, trained personnel, and transport costs may be beyond the resources of veterinary services. In addition, transport of samples from the point of collection to the laboratory can be often impeded due to poor infrastructure, further delaying EHD laboratory confirmation. The development of more affordable assays, suitable for field deployment, would improve the control of a potential EHDV outbreak or/and support any surveillance programs in resource-poor settings.

Reverse transcription loop-mediated isothermal amplification (RT-LAMP) is a rapid, highly sensitive nucleic acid amplification technique that could be deployed in the field or used for high throughput screening in diagnostic laboratories. RT-LAMP utilizes a set of four to six primer pairs, which target six to eight regions, increasing the assay specificity. The strand-displacing polymerases used in RT-LAMP assays are more robust and less affected by PCR inhibitors, enabling simplification of the extraction procedures [21,22,23,24]. As RT-LAMP is performed at a constant temperature, it does not require an expensive thermal cycler; therefore, positive results can be observed though color change dyes or the development of turbidity [24]. RT-LAMP assays can also be monitored in real-time through the increase in the fluorescence of a double-stranded DNA (dsDNA) binding dye [22,23], using either portable instruments or more expensive real-time PCR instruments. Several (RT-) LAMP assays have been developed for the detection of livestock viruses such as BTV [25], African horse sickness virus [26], Peste des petits ruminants virus [22], and foot-and-mouth virus [23]. In human medicine, RT-LAMP has recently gained popularity as a rapid and accurate CE-IVD marked test for SARS-CoV-2 detection within UK laboratories [27,28].

In this study, we designed two sets of LAMP primers allowing for the rapid detection of eastern and western EHDV topotypes, to be used either in a singleplex or duplex RT-LAMP format. The RT-LAMP assay formats (eastern, western, and multiplex) were evaluated in comparison with a “gold standard” RT-qPCR assay using 43 cell culture isolates representing geographically distinct EHDV strains (eastern and western), field specimens from Kenya, and an EHDV proficiency testing panel.

## 2. Materials and Methods

### 2.1. Primer Design

To search for conserved regions (or segments) across the EHDV genome, a multiple sequence alignment of 54 full-length genomes was initially performed in MEGA6 [29]. For the primer design, a sequence each for serotypes 1, 2, 4, 5, 6, 7, 8, and 10 (Genbank accession numbers AM744977, AM744987, AM745017, AM745027, AM745037, AM745047, AM745057, and LC202944, respectively) were aligned using clustalX [30] and a consensus sequence made using Clustal Consensus Maker (GeneSys Biotech Ltd., Camberley, UK). Conserved regions were selected by eye and primers were designed in these regions using LAMP Designer (PREMIER Biosoft, San Francisco, CA, USA) against individual sequences. Primer sets, which targeted the most conserved regions when compared with the clustalX alignment, were selected for testing. For the selected primer sets (Table 1), assay conditions were initially optimized with regards to the following: master mix selection (OptiGene Limited reagents: ISO-001 + RT vs. ISO-004 + RT) and primer concentrations.

For the multiplex assay, optimization of the primer combinations was performed by adding s9.E primers into the s9.W assay. The following combinations were tested: (i) s9.W plus s9.E FIP/BIP, (ii) s9.W plus s9.E FIP/BIP/LoopF, (iii) s9.W plus s9.E FIP/BIP/LoopB, (iv) s9.W plus s9.E FIP/BIP/F3, and (v) s9.W plus s9.E FIP/BIP/B3.

### 2.2. RNA Extraction

Automated extraction of the EHDV RNA was performed using 100 µL of cell culture isolates representing all of the EHDV serotypes (EHDV-1, -2, -4, -5, -6, -7, -8, and -10) on the Kingfisher Flex automated extraction platform (ThermoFisher Scientific, Paisley, UK) and the MagVet Universal nucleic acid extraction kit (ThermoFisher). The RNA was eluted into 80 µL of elution buffer and was stored at 4 °C prior to analysis using the RT-qPCR assay or the RT-LAMP assay.

### 2.3. RT-LAMP

EHDV RNA was denatured at 95 °C for 5 min using a *Veriti* 96 Well Fast Thermal Cycler (Applied Biosystems, UK) and was placed on a cool block before being used as the template. The RT-LAMP assay was performed using the RT Isothermal Master Mix (OptiGene Ltd., Horsham, UK) in a 25 µL total reaction mixture volume containing 5 µL of template, 15 µL of RT isothermal master mix ISO-004-RT300 (OptiGene Ltd., UK), 2.5 µL 10 × primer mix (Table 1), and 2.5 µL nuclease-free water. RT-LAMP was performed at 65 °C for 20 min using the Genie^®^ II portable fluorimeter (OpiGene Ltd.). Anneal analysis was performed by heating the reaction to 98 °C for 1 min, then cooling to 80 °C, and decreasing at 0.05 °C/s to confirm that the amplicons were EHDV-specific. For larger quantities of samples (>30), RT-LAMP was performed on an Applied Biosystems 7500 Fast instrument (Life Technologies, Paisley, UK) as OptiGene reagents utilize a fluorescent dsDNA intercalating dye that can be detected in the SYBR green channel.

### 2.4. RT-qPCR

EHDV RNA (3.4 µL) was denatured at 95 ° for 5 min using a Veriti 96 Well Fast ThermalCycler (Applied Biosystems, Birchwood, UK) and placed on cool block. The Maan et al. (2017) [19] assay was performed using 20 µL of the reaction mix comprising the Express One-Step Superscript qRT-PCR kit (LifeTechnologies, Paisley, UK) containing 1 × reaction mix, 800 nM forward and reverse primers, 100 nM probe, 0.4 µL Rox, and 2 µL of enzyme mix in each well. RT-qPCR was performed on an Applied Biosystems 7500 Fast instrument (Life Technologies) with the following cycling conditions: reverse transcription at 50 °C for 15 min, RT inactivation/Taq activation at 95 °C for 20 s, and then 45 cycles of PCR, with each cycle consisting of 95 °C for 3 s and 60 °C for 30 s.

### 2.5. Validation of RT-LAMPs

#### 2.5.1. In-House Specificity

Analytical specificity was determined across 43 cell culture isolates, representing different EHDV serotypes and topotypes collected across the world. Among the EHDV isolates tested were EHDV-6 isolates originating from the countries neighboring the European Union (EU) such as Turkey, Morocco, Algeria, and Israel, as well as EHDV-1, -2, and -6 isolates representing the current EHDV serotypes circulating in the United States of America. All isolates were obtained from the Orbivirus reference collection (ORC) at The Pirbright Institute, full details of these viruses are available on the dsRNA virus website (https://www.reoviridae.org/ accessed on 29 August 2021). The specificity of the RT-LAMP assay was determined using previously extracted RNA derived from BTV and foot-and-mouth disease virus (FMDV) isolates.

#### 2.5.2. Analytical Sensitivity and Limit of Detection

Two of the most recent EHDV isolates, representing western (TUR2007/01) and eastern (TAT2013/02) topotypes, were selected to determine the analytical sensitivity of the RT-LAMP assays: s9.E, s9.W, and multiplex. A standard curve was generated using log dilution (10^−1^ to 10^−4^), and each dilution was tested in duplicate using the RT-qPCR and the RT-LAMP assay. Then, a one in two dilution series was generated from the last detectable dilution (10^−3^ or 10^−4^) and was tested in quadruplet to determine the limit of detection (LOD), which was considered as the greatest dilution for which all replicates tested positive. For each dilution, the percentage coefficient of variation (%CV) was calculated to assess the repeatability of the assay throughout the detection range.

#### 2.5.3. Field Samples

Blood and tissue samples (n = 300) were collected as part of the Infectious Diseases in East African Livestock (IDEAL) project, and were tested using a range of diagnostic methods for over 100 different pathogens. The IDEAL project monitored infections in 548 indigenous calves from birth to death or 12 months of age in Western Kenya between 2007–2009 [31]. A subset of EHDV plausible positive samples (n = 20) was analyzed using the RT-qPCR and multiplex RT-LAMP assays in the OIE reference laboratory for Bluetongue virus at the Pirbright Institute. All EHDV positive samples were thought to belong to the western topotype and were also tested using RT-LAMP assay with the s9.W primer set.

#### 2.5.4. Proficiency Panel

An EHDV real time RT-qPCR panel (n = 10) was kindly supplied by the National Center for foreign Animal Disease (NCFAD) of the Canadian Food Inspection Agency, Winnipeg, Manitoba. Each sample was tested in duplicate using both s9.E and s9.W RT-LAMP assays, and these results were compared to the RT-qPCR generated by NCFAD.

## 3. Results

### 3.1. Assay Design and Primer Evaluation

Using LAMP Designer, the primers could not be designed for segments 2, 3, 6, 7, 8, and 10, due to their high sequence variability. For the remaining segments (1, 4, 5, and 9), primer sets were designed and initially evaluated. Of the four target regions, primers targeting segment 9 displayed the fastest amplification time and were taken forward for further testing (data not shown). It was not possible to design one primer set for the detection of all-known EHDV strains due to several mismatches in the alignment; therefore, two separate primer sets targeting eastern (s9.E) and western (s9.W) topotypes were designed and evaluated in this study in two formats: singleplex and multiplex.

For the multiplex assay, a variety of primer combinations were tested on two EHDV serotypes (Table 2). Combining primers s9.W with the FIP (2 µM), BIP (2 µM), and LoopB (1 µM) primer from s9.E resulted in the fastest amplification time. The concentrations of these primers were then optimized (data not shown) (final concentrations shown in Table 1) and taken forward for further testing.

To eliminate the risk of false-positives, primer sets s9.E, s9.W, and multiplex primer were tested on EHDV-negative samples, such as EDTA blood (n = 10), spleen (n = 20), and brain (n = 2) originating from bovine and ovine species. None of the primer sets (s9.E, s9.W, and multiplex) gave non-specific amplification even when the initial duration of the LAMP was extended to 30 min.

### 3.2. Validation of RT-LAMPs

#### 3.2.1. In-House Specificity

None of the RT-LAMP primer sets (s9.E, s9.W, and multiplex) displayed cross-reactivity with other viruses such as BTV (serotypes -1, -2, -4, -8, -9, and -16) or FMDV (SAT2). Of the 43 EHDV isolates tested, all of the eastern topotypes (n = 11) were detected using s9.E primer set, and all of the western topotypes (n = 32) were detected using the s9.W primer set (Table 3). The multiplex RT-LAMP assay detected all EHDV isolates (n = 43), irrespective of the topotype. Interestingly, one isolate NIG1967/01 was negative using the EHDV RT-qPCR, but positive using the RT-LAMP assay (s9.W and multiplex primer sets). RT-qPCR testing of NIG1967/01 was repeated twice, but negative results were still obtained. Then, an aliquot from our long-term liquid nitrogen stock was requested for RT-qPCR testing and a weak positive was obtained, whereas a strong positive result was observed with the RT-LAMP assay (s9.W and multiplex primer sets).

Based on the 43 EHDV isolates tested, the mean anneal temperatures (T_a_) were 87.05 °C ± 0.32 for set s9.E across the 11 eastern topotypes, 86.75 °C ± 0.38 for set s9.W across the 32 western topotypes, and 87.06 °C ± 0.51 for the multiplex assay across all samples.

#### 3.2.2. Sensitivity and Limit of Detection

The relationship between the mean t_p_ value and the mean C_T_ value was linear, with an R-squared value (R^2^) of 0.95 and 0.93 for TAT2013/02 eastern and TUR2007/01 western topotype isolates, respectively. An acceptable %CV for diagnostic assays is thought to be 10% [32]; however, greater %CVs values such as 22.97% (eastern topotype) and 40.05% (western topotype) were generated for the last detectable dilution for both RT-LAMP assays (Table 4 and Table 5), indicating that the intra-assay repeatability decreases when approaching the LOD. The LOD of western, eastern, and multiplex RT-LAMP assays was estimated in comparison with the C_T_ values of the RT-qPCR assay. The LOD of eastern and western RT-LAMP assays was estimated as ~24.36 C_T_ and ~29.37 C_T_, respectively (Table 3 and Table 4), indicating a greater sensitivity of the western topotype singleplex RT-LAMP assay. The multiplex RT-LAMP assay was equally sensitive towards the eastern topotypes as the eastern-topotype singleplex RT-LAMP assay (~24.36). However, the multiplex RT-LAMP assay was less sensitive towards western topotypes in comparison with the western singleplex RT-LAMP assay, and its sensitivity towards TUR2007/01 dropped from ~29.37 to ~27.53 C_T_.

#### 3.2.3. Field Samples

The sensitivities of the RT-LAMP s9.W assay and the multiplex RT-LAMP assay, relative to the RT-qPCR assay, were 72.2% and 33.3%, respectively (Table 6). Sample W19/17 206 was negative using the RT-qPCR assay, but positive using the RT-LAMP s9.W assay, and the positive and negative controls used during the analysis did not indicate cross-contamination. The multiplex LAMP assay showed a drop of sensitivity in relation to the RT-LAMP s9.W assay by failing to detect 53.8% of the EHDV positive samples.

There was a slight increase in the mean T_a_ when testing the blood and tissue samples (87.04 °C ± 0.22) in comparison with the cell culture isolates (86.75 °C ± 0.38) using the s9.W RT-LAMP assay. A similar increase in the mean T_a_ was observed for the multiplex LAMP assay (87.06 °C ± 0.51 for blood and tissue samples, and 87.12 °C ± 0.29 for cell culture isolates).

#### 3.2.4. Proficiency Panel

Six out of ten samples were EHDV positive using the RT-qPCR assay performed in the NCFAD, Canadian Food Inspection Agency, Winnipeg, Manitoba. At the Pirbright Institute, all of the samples were tested negative using the s9.E RT-LAMP assay and three were positive using the s9.W RT-LAMP assay, indicating their western topotype origin (Table 7). Three RT-qPCR EHDV positive samples of C_T_ greater than 34.00 were detected by neither the s9.E RT-LAMP nor the s9.W RT-LAMP assays.

## 4. Discussion

For the first time, we developed EHDV RT-LAMP assays that enable rapid, low-cost, and accurate detection of EHDV and that have the potential to be used in the field. Due to the considerable sequence diversity of EHDV, it was not possible to design a single set of LAMP primers capable of detecting all of the EHDV strains. Therefore, we designed two separate sets of primers for the detection of western (s9.W) and eastern (s9.E) topotype strains, both targeting the Seg-9 of the EHDV genome. Both primer sets demonstrated a high specificity towards EHDV by detecting over 40 EHDV isolates representing eight known serotypes, originating from different locations worldwide. The RT-LAMP assays can be used in either in singleplex and multiplex format. Although the singleplex format is more sensitive, the multiplex RT-LAMP assay could be employed in areas where co-circulation of both eastern and western strains is suspected. In addition, the RT-LAMP assays could be used to rapidly confirm/rule out suspicion of EHD in a BTV-free area.

Eastern strains are considered less pathogenic than western strains [8] as they were primarily detected in wild-caught *Culicoides* or asymptomatic sentinel cattle [33]. In general, there are more reports of EHD following infection with western strains (e.g., American, Africa, and the Middle-East) than eastern strains, with the exception of Japan, where eastern strains of EHDV-2 (e.g., JAP1959/01) [4], EHDV-6 (e.g., HG-1/E/15) [34], and EHDV-7 [35] have caused a significant disease in cattle on multiple occasions. In this study, the sensitivity of the s9.W RT-LAMP assay was calculated as 72.2% in relation to RT-qPCR assay when the blood and tissue samples from Kenyan calves were analyzed. Some samples were tested as negative for EHDV using the RT-qPCR, despite being recorded as being positive previously. These field samples were collected in a cohort study between 2007 and 2009 in Kenya, and were stored prior to analysis; therefore, it cannot be ruled out that freeze–thaw cycles could impact on the quality and integrity of RNA for diagnostic testing. It would be interesting to compare the performance of both RT-qPCR and RT-LAMP assays on specimens collected from animals exhibiting clinical signs of EHD during outbreaks.

The LOD of s9.W RT-LAMP assay was estimated at approximately 29.37 C_T_ value, indicating that this assay could be useful to detect EHDV infected animals during peak viraemia or late viraemia. At peak viraemia (4–9 day post infection (dpi)), mean C_T_ levels of 25.65 (range from 21.18 to 29.33) were detected in ten experimentally EHDV-6 infected Holstein-Friesian cattle [36], indicating that they could be theoretically detected by the s9.W RT-LAMP assay. Similarly, monthly sampling of dairy cattle showed that C_T_ values ranged from 22.30 to 35.40 in the first-time EHDV infected animals in Trinidad and Tobago [15]. In addition, blood viral titers in white-tailed deer, exhibiting mild and moderate clinical signs, ranged from 10^3.7^ to 10^6^ TCID_50_/mL [37] at peak viraemia (5 dpi), equating roughly to C_T_ values of 20.87 to 28.88. In another study, blood viral titers of 10^4.6^ to 10^5.26^ TCID_50_ (~C_T_ 23.44 to 25.75) were detected at 8 dpi when the majority of infected fawns shown clinical signs of EHD [38]. These studies suggest that samples obtained from viraemic bovines and cervines could be rapidly detected as they would fall within the detection range of the s9.W RT-LAMP assay. In contrast, the s9.E RT-LAMP assay (LOD of ~24.36 C_T_) was less sensitive in comparison with the western topotype RT-LAMP assay. However, the sensitivity of the s9.E RT-LAMP may be sufficient to be used as a rapid diagnostic tool in the field when several animals from the same herd, likely at different stages of infection, are tested.

We were able to easily adapt the RT-LAMP assay for use in our laboratory by utilizing equipment, plasticware, and laboratory set-ups routinely used for real-time RT-qPCR diagnostics. As OptiGene reagents contain a fluorescent dsDNA intercalating dye detectable by the AB7500 Fast instrument (SYBR green/FAM channel), it was possible to use this equipment after adjusting the cycling conditions and melting curve settings. In addition, the Applied Biosystems 7500 Fast instrument can simultaneously analyze 96 reactions in one optical 96-well plate, in comparison with 16 reactions per run offered by the Genie II instrument. Furthermore, when using the AB7500 fast instrument, the estimated cost per reaction was £2.40 for the RT-LAMP and £4.80 for the RT-qPCR assay, whereas the reaction time was at least four times shorter for the RT-LAMP assay. With the current interest in the use of LAMP technology for rapid detection of emerging human viruses such as SARS-CoV-2 [39] or Zika virus [40], the upscaling of RT-LAMP for high throughput screening in diagnostic laboratories could make this technology a good alternative to RT-qPCR, especially during a shortage of RT-qPCR diagnostic reagents, as noted during the COVID-19 pandemic, but, crucially, when a cheap but fast assay is preferable.

The RT-LAMP assays developed in this study could be further adapted for point of use application; however, some future work is required and should consider either the removal of nucleic extraction steps [39] or the use of field-suitable extraction procedures [41]. To help reduce the cost of in-field diagnostics, the development of an alternative visualization strategy, such as a direct colorimetric LAMP assay [42], could be considered. Ideally, to implement these improvements, the RT-LAMP assay validation should be performed directly in the field during outbreak investigations.

## 5. Conclusions

In conclusion, we have developed novel, rapid EHDV RT-LAMP assays targeting western and eastern topotype strains, available either in singleplex and multiplex format. The s9.W RT-LAMP assay had sufficient sensitivity and specificity, and could be used to detect more pathogenic strains (western topotype) either in established laboratories or in the field so as to control the spread of EHDV.

## Figures and Tables

**Table 1 viruses-13-02187-t001:** RT-LAMP primer sets.

Set	Primer ^1^	Sequence (5′-3′)	Final Reaction Concentration (µM)
Eastern topotype s9.E	F3	GACGCCTGGATGTTACAG	0.2
	B3	GCAGCGACTTCTCAATGT	0.2
	FIP (F1c + F2)	CTTCCAGTTCCTGACGCATCATATCTAGCGACGGAGGAG	2
	BIP (B1c + B2)	AATAGAGGGAGATGGGTAGTGGGTGCTCACTCCGTACCG	2
	LoopF	CTCTATCTCCTCTCTTAGTCTCACT	1
	LoopB	GAGTGAAGAAATCGCTCAATGTC	1
Western topotype s9.W	F3	GATGTTCGACGCATGGAT	0.2
	B3	CGTACCATTTGCTCCAGG	0.2
	FIP (F1c + F2)	TCCAGTTCTTGCCGCATCATTGATCTGGAGAACGCAAGG	2
	BIP (B1c + B2)	ACGGGAGGAGGAAGATGGTTGGAATACTCACTCCGTACCTA	2
	LoopF	TCTATCTCCTCCCTTAACCTTA	1
	LoopB	GAGCGAAGAGATAGCACAAT	1
Multiplex	F3	GATGTTCGACGCATGGAT	0.2
	B3	CGTACCATTTGCTCCAGG	0.2
	FIP (F1c + F2)	TCCAGTTCTTGCCGCATCATTGATCTGGAGAACGCAAGG	1.6
	BIP (B1c + B2)	ACGGGAGGAGGAAGATGGTTGGAATACTCACTCCGTACCTA	1.6
	LoopF	TCTATCTCCTCCCTTAACCTTA	0.8
	LoopB	GAGCGAAGAGATAGCACAAT	0.8
	FIP (F1c + F2)	CTTCCAGTTCCTGACGCATCATATCTAGCGACGGAGGAG	1.6
	BIP (B1c + B2)	AATAGAGGGAGATGGGTAGTGGGTGCTCACTCCGTACCG	1.6
	LoopB	GAGTGAAGAAATCGCTCAATGTC	0.8

^1^ Primers were obtained from ThermFisher Scientific.

**Table 2 viruses-13-02187-t002:** Performance of different primer combinations during the multiplex RT-LAMP assay validation.

Sample ID	RT-qPCR C_T_ Value	s9.W	s9.W + s9.E FIP, BIP	s9.W + s9.E FIP, BIP, LoopF	s9.W + s9.E FIP, BIP, LoopB	s9.W + s9.E FIP, BIP, F3	s9.W + s9.E FIP, BIP, B3
		RT-LAMP t_p_ [min] (T_a_ [°C])					
EHDV-1	20.03	7.50 (86.40)	8.50 (86.45)	10.25 (86.41)	10.75 (86.41)	10.75 (86.41)	10.25 (86.41)
EHDV-8	19.72	ND	12.25 (87.20)	11.75 (87.15)	8.75 (87.15)	12.25 (87.20)	ND

ND—not detected. Performed using ISO-001 mastermix and a 20-min cut-off time. Primer concentrations and mastermix type (ISO-004) were later optimized.

**Table 3 viruses-13-02187-t003:** EHDV isolates tested during the development of the RT-LAMP assays.

Sample ID	Serotype (Topotype)	RT-qPCR C_T_ Value	RT-LAMP s9.E		RT-LAMP s9.W		RT-LAMP Multiplex	
			t_p_ [min]	T_a_ [°C]	t_p_ [min]	T_a_ [°C]	t_p_ [min]	T_a_ [°C]
AUS1977/01	EHDV-5 (eastern)	15.02	5.37	86.99	ND	ND	4.94	87.44
AUS1980/03	EHDV-2 (eastern)	15.58	5.67	87.44	ND	ND	9.45	87.29
AUS1981/06	EHDV-7 (eastern)	11.11	6.52	86.55	ND	ND	5.05	86.84
AUS1981/07	EHDV-6 (eastern)	18.76	3.90	86.99	ND	ND	5.70	87.59
AUS1982/05	EHDV-8 (eastern)	14.64	5.14	87.29	ND	ND	9.60	87.14
AUS1995/02	EHDV-1 (eastern)	12.52	5.62	87.14	ND	ND	10.48	87.74
ISA1988/01	EHDV-2 (eastern)	14.41	6.90	87.33	ND	ND	7.43	87.48
ISA1990/01	EHDV-2 (eastern)	14.43	6.00	87.18	ND	ND	7.44	87.63
ISA1991/03	EHDV-10 (eastern)	12.27	6.10	86.40	ND	ND	7.00	86.55
JAP1959/01	EHDV-2 (eastern)	17.12	5.86	87.14	ND	ND	7.03	87.59
TAT2013/02	EHDV-6 (eastern)	15.25	4.24	87.14	ND	ND	5.26	87.74
ALG2006/02	EHDV-6 (western)	14.66	ND	ND	4.33	86.84	5.51	86.84
BAR1983/01	EHDV-6 (western)	13.82	ND	ND	4.23	87.29	4.53	87.59
BRA2008/01	EHDV-2 (western)	14.88	ND	ND	8.65	86.40	10.96	86.69
CAN1962/01	EHDV-2 (western)	13.90	ND	ND	4.77	86.55	5.54	86.99
GLP2011/01	EHDV-2 (western)	11.86	ND	ND	5.23	86.55	8.24	87.29
GUI2011/01	EHDV-1 (western)	14.54	ND	ND	7.88	86.99	8.66	87.59
ISR2006/01	EHDV-7 (western)	13.12	ND	ND	5.47	86.88	6.22	87.03
MOR2004/03	EHDV-7 (western)	15.05	ND	ND	4.70	86.84	6.66	86.84
MOR2006/05	EHDV-6 (western)	15.42	ND	ND	4.67	86.99	4.87	87.74
NIG1967/01	EHDV-1 (western)	ND	ND	ND	5.39	87.29	11.98	87.59
NIG1968/01	EHDV-4 (western)	17.54	ND	ND	4.60	87.44	5.36	87.88
REU2003/03	EHDV-6 (western)	16.86	ND	ND	4.66	86.88	4.46	86.88
REU2009/01	EHDV-6 (western)	13.31	ND	ND	5.09	87.29	5.68	87.29
SND1982/04	EHDV-6 (western)	12.08	ND	ND	4.22	87.03	4.72	87.33
SND1982/05	EHDV-6 (western)	13.84	ND	ND	4.14	87.03	5.09	87.33
SND1983/02	EHDV-6 (western)	13.76	ND	ND	3.81	87.03	4.76	87.18
TUR2007/01	EHDV-6 (western)	13.92	ND	ND	5.17	86.99	5.64	86.99
USA1955/01	EHDV-1 (western)	19.55	ND	ND	7.32	86.69	7.67	86.69
USA1978/01	EHDV-2 (western)	15.93	ND	ND	6.00	86.55	5.80	86.99
USA1980/01	EHDV-2 (western)	15.18	ND	ND	5.06	86.40	6.26	86.55
USA1993/01	EHDV-2 (western)	13.26	ND	ND	6.59	86.25	6.47	86.84
USA1994/01	EHDV-2 (western)	14.25	ND	ND	5.52	86.69	10.86	87.14
USA1996/01	EHDV-2 (western)	12.89	ND	ND	6.20	86.69	8.03	86.69
USA1996/02	EHDV-2 (western)	13.69	ND	ND	5.57	86.55	6.05	86.99
USA1996/04	EHDV-1 (western)	9.64	ND	ND	5.21	86.10	7.74	85.80
USA1998/01	EHDV-2 (western)	11.10	ND	ND	4.93	86.69	6.09	86.55
USA1999/01	EHDV-2 (western)	12.87	ND	ND	6.64	86.84	7.89	87.14
USA1999/04	EHDV-1 (western)	12.73	ND	ND	5.13	86.10	8.02	85.95
USA2000/01	EHDV-2 (western)	14.38	ND	ND	5.95	87.14	7.67	87.14
USA2001/01	EHDV-1 (western)	13.93	ND	ND	4.73	86.69	11.89	86.69
USA2001/07	EHDV-2 (western)	12.45	ND	ND	4.93	85.95	7.38	85.65
USA2006/05	EHDV-6 (western)	13.00	ND	ND	6.00	86.25	6.18	86.69

ND—not detected.

**Table 4 viruses-13-02187-t004:** Intra-assay repeatability of the EHDV RT-LAMP for primer set s9.E.

Sample ID	Dilution	Mean C_T_ Value	Mean T_a_ [°C]	Mean tp [min]	Standard Deviation	%CV
TAT2013/02	10^−1^	15.00	87.21	3.77	0.01	0.22
	10^−2^	18.77	87.36	4.63	0.02	0.46
	1 in 2 of 10^−2^	20.38	87.21	5.64	1.05	18.59
	1 in 4 of 10^−2^	22.02	87.51	7.04	0.37	5.30
	1 in 8 of 10^−2^	23.53	87.36	7.12	1.11	15.54
	1 in 16 of 10^−2^	24.36	87.07	7.73	1.78	22.97

**Table 5 viruses-13-02187-t005:** Intra-assay repeatability of the EHDV RT-LAMP for primer set s9.W.

Sample ID	Dilution	Mean C_T_ Value	Mean T_a_ [°C]	Mean tp [min]	Standard Deviation	%CV
TUR2007/01	10^−1^	16.27	87.63	3.96	0.02	0.55
	10^−2^	20.15	87.48	4.79	0.01	0.26
	10^−3^	24.07	87.25	6.45	0.47	7.22
	1 in 2 of 10^−3^	26.19	87.36	6.24	0.63	10.17
	1 in 4 of 10^−3^	27.51	87.66	7.45	1.33	17.80
	1 in 8 of 10^−3^	29.37	87.36	12.86	5.15	40.05

**Table 6 viruses-13-02187-t006:** The RT-qPCR and the RT-LAMP (singleton s9.W, multiplex) results in the field samples.

Sample ID	RT-qPCR C_T_ Value	RT-LAMP s9.W		RT-LAMP Multiplex	
		t_p_ [min]	T_a_ [°C]	t_p_ [min]	T_a_ [°C]
W19/17 104	31.13	ND	ND	ND	ND
W19/17 109	24.56	5.20	87.14	7.82	87.44
W19/17 129	29.10	5.67	87.14	14.34	87.29
W19/17 133	26.87	9.41	87.14	8.97	87.29
W19/17 138	28.01	15.12	86.99	ND	ND
W19/17 140	31.45	7.02	87.14	ND	ND
W19/17 145	24.58	7.79	86.99	9.69	86.84
W19/17 165	28.89	13.63	86.69	9.14	86.69
W19/17 169	31.82	ND	ND	ND	ND
W19/17 175	30.48	9.68	86.99	ND	ND
W19/17 186	ND	ND	ND	ND	ND
W19/17 192	ND	ND	ND	ND	ND
W19/17 195	26.80	9.23	87.29	10.18	87.14
W19/17 206	ND	14.98	86.84	ND	ND
W19/17 211	28.29	12.17	86.69	ND	ND
W19/17 226	27.12	ND	ND	ND	ND
W19/17 233	26.64	8.98	86.99	ND	ND
W19/17 260	27.39	ND	ND	ND	ND
W19/17 275	28.17	ND	ND	ND	ND
W19/17 277	28.97	5.93	87.44	ND	ND

ND—not detected.

**Table 7 viruses-13-02187-t007:** The RT-qPCR and the RT-LAMP (singleton s9.E, s9.W) results in the proficiency panel.

PCR Panel	RT-qPCR C_T_ Value	RT-LAMP s9.E		RT-LAMP s9.W	
		t_p_ [min]	T_a_ [°C]	t_p_ [min]	T_a_ [°C]
S01	17.51	ND	ND	4.41	86.84
		ND	ND	4.72	86.99
S02	ND	ND	ND	ND	ND
		ND	ND	ND	ND
S03	19.12	ND	ND	3.50	86.69
		ND	ND	3.56	86.69
S04	ND	ND	ND	ND	ND
		ND	ND	ND	ND
S05	34.25	ND	ND	ND	ND
		ND	ND	ND	ND
S06	ND	ND	ND	ND	ND
		ND	ND	ND	ND
S07	18.63	ND	ND	6.93	87.14
		ND	ND	6.79	86.99
S08	ND	ND	ND	ND	ND
		ND	ND	ND	ND
S09	35.17	ND	ND	ND	ND
		ND	ND	ND	ND
S10	35.22	ND	ND	ND	ND
		ND	ND	ND	ND

ND—not detected.

## Data Availability

Data and related research documentation are archived at The Pirbright Institute.

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
