# Peer review of "Development of a Novel Loop Mediated Isothermal Amplification Assay (LAMP) for the Rapid Detection of Epizootic Haemorrhagic Disease Virus"

_viruses, 2021, doi:10.3390/v13112187_

Round 1

Reviewer 1 Report

The study is well-designed and the manuscript is well written. 

The main criticism is that in the introduction the authors say that RT-LAMP assays can be run without expensive equipment to provide a faster, lower cost alternative to submitting samples to national labs in countries with fewer resources. However, the authors validate their assay using the very equipment they say this assay does not require. This study would have been much more impressive if the authors had validated the assay for field use or in resource poor settings.

Author Response

Point 1: The main criticism is that in the introduction the authors say that RT-LAMP assays can be run without expensive equipment to provide a faster, lower cost alternative to submitting samples to national labs in countries with fewer resources. However, the authors validate their assay using the very equipment they say this assay does not require. This study would have been much more impressive if the authors had validated the assay for field use or in resource poor settings.

Response 1:

Thank you very much for this valid comment-we agree with the reviewer in this regard. As RT-LAMP assays do not require expensive equipment (as a water bath and the portable Genie II instrument is sufficient to perform analysis) they can be adapted to field diagnostics. It was unfortunate that we did not have enough funding to performing subsequent epidemiological field studies. Notwithstanding, we designed an RT-LAMP assay for the detection of EHDV utilising 43 cell culture isolates (representing a unique global collection of EHDV serotypes and topotypes). We consider this RT-LAMP assay as being well-placed for use in the field at a later stage. In our opinion (given the instrumentation we utilised herein), the assay is sensitive and specific to detect EHDV and could be used to analyse historical sample sets since it offers a much cheaper alternative to PCR or indeed sequencing.

Reviewer 2 Report

Over all good paper, some minor typos noted such as:

line 65  - as for with BTV, EHDV....

line 260 - negative is spelt incorrectly

line 336 - the number 4 should be written out since there is no unit. Any number between 0-10 when there is no unit is written.

Author Response

Point1: line 65  - as for with BTV, EHDV....

Response 1: Corrected as recommended

Point2: line 260 - negative is spelt incorrectly

Response 2: Spelling was corrected

Point3: line 336 - the number 4 should be written out since there is no unit. Any number between 0-10 when there is no unit is written.

Response 3: Corrected as recommended